# Contrast-Enhanced Harmonic Endoscopic Ultrasound-Guided Puncture for the Patients with Pancreatic Masses

**DOI:** 10.3390/diagnostics13061039

**Published:** 2023-03-08

**Authors:** Yasuo Otsuka, Ken Kamata, Masatoshi Kudo

**Affiliations:** Department of Gastroenterology and Hepatology, Kindai University Faculty of Medicine, Osakasayama 589-8511, Japan; yotsuka624@gmail.com (Y.O.);

**Keywords:** avascular, contrast, endoscopic ultrasonography, endoscopic ultrasound-guided fine-needle aspiration, pancreatic cancer

## Abstract

Endoscopic ultrasound-guided fine-needle aspiration (EUS-FNA) is useful for the diagnosis of pancreatic masses. According to three meta-analyses, the sensitivity, specificity, and accuracy of EUS-FNA are 84–92%, 96–98%, and 86–91%, respectively. However, the occurrence of false-negative and false-positive results indicates that the diagnostic performance of EUS-FNA needs to be improved. Contrast-enhanced harmonic endoscopic ultrasonography (CH-EUS) is used for the characterization of pancreatic masses and can be applied to improve the performance of EUS-FNA. When CH-EUS is used to evaluate intratumor blood flow, an avascular area inside the pancreatic mass that is considered to be fibrosis is often detected. This area can be avoided by performing EUS-FNA under CH-EUS guidance. In this review, we summarize the data on contrast-enhanced harmonic endoscopic ultrasound-guided fine-needle aspiration (CH-EUS-FNA), which suggest that its benefit is still a matter of debate. Of eight studies analyzed, only one showed that CH-EUS improved the sensitivity of EUS-FNA. The future challenge is to determine under what circumstances CH-EUS-FNA is useful.

## 1. Introduction

Endoscopic ultrasonography (EUS) allows detailed visualization of the pancreas and the localization of pancreatic solid masses. Endoscopic ultrasound-guided fine-needle aspiration (EUS-FNA) was first applied clinically by Vilmann et al. in 1992 [1], and is currently widely used for the pathological diagnosis of pancreatic solid masses. According to three meta-analyses evaluating the diagnostic performance of EUS-FNA for pancreatic masses, its sensitivity, specificity, and accuracy range between 84–92%, 96–98%, and 86–91%, respectively [2]. Thus, EUS-FNA is associated with a few false-negative and false-positive results. Contrast-enhanced harmonic endoscopic ultrasonography (CH-EUS) allows the visualization of intratumor blood flow using an ultrasound contrast agent, such as Perflubutane microspheres, and is applied for the identification and characterization of pancreatobiliary masses [3,4]. Although EUS-FNA is usually performed under EUS guidance, CH-EUS can be used to guide the needle to a specific site in the tumor to improve specimen collection. In this review, we summarize the literature on the methods and implications of contrast-enhanced harmonic endoscopic ultrasound-guided fine-needle aspiration (CH-EUS-FNA), and discuss the potential of CH-EUS-FNA for improving the diagnostic performance of EUS-FNA in patients with pancreatic cancer. In recent years, the efficacy of ultrasound contrast agents has been reported in regards to procedures such as endoscopic ultrasound-guided biliary drainage (EUS-BD) utilizing EUS. The effectiveness of ultrasound contrast agents in EUS-related procedures will also be reviewed.

## 2. The CH-EUS-FNA Technique

In selecting the literature, the following search terms were used in PubMed: contrast (title or abstract) OR contrast-enhanced (title or abstract) OR contrast-enhanced harmonic (title or abstract) OR CE-EUS (title or abstract) OR CH-EUS (title or abstract) OR CEH-EUS (title or abstract) AND endoscopic ultrasound (title or abstract) OR EUS (title or abstract) OR endosonography (title or abstract or MeSH terms) OR endoscopic ultrasonography (title or abstract) AND FNA (title or abstract) OR FNB (title or abstract) OR fine needle aspiration (title or abstract) OR fine needle biopsy (title or abstract) OR sampling (title or abstract). Then, after sequential screening of abstracts and texts, eight studies were determined as shown in Table 1 [5,6,7,8,9,10,11,12]. In most studies, CH-EUS-FNA was performed in the late phase of CH-EUS (Table 1), suggesting that CH-EUS-FNA was performed after evaluation of blood flow in the pancreatic mass in late-phase CH-EUS. However, in one study, CH-EUS-FNA was performed in the early arterial phase [12].

A prolonged contrast period is important for EUS-FNA, which normally requires more than two passes. Second-generation ultrasound contrast agents such as Sulphur hexafluoride microbubbles, Perflutren lipid microspheres, and Perflubutane microspheres resonate under low acoustic power and generate a second harmonic component, which provides at least several minutes of contrast effect [13,14]. Unlike other contrast medias, perflubutane microspheres have the advantage of obtaining a Kupffer image. Perflubutane microspheres allow contrast-enhanced ultrasound evaluations at early phase, late phase, and Kupffer phase. The early, late, and Kupffer phases are defined as 10–30 s, 30–120 s, and 10 min after injection of the contrast agent, respectively [15]. There are no Kupffer cells in the pancreas; therefore, early and late phases are used for CH-EUS evaluations for pancreatic lesions and the significance of this advantage in the diagnosis of pancreatic tumors is not presently known. Thus, any second-generation ultrasound contrast agents can be used for the diagnosis of pancreatic tumors. In EUS-FNA, the fanning technique (sampling multiple areas with each needle pass) is recommended to obtain tumor tissue from a hot spot [16]. However, CH-EUS-FNA has the advantage that any avascular area can be avoided and the fanning technique is not always applicable. It remains unclear whether the early or late phase of CH-EUS is more appropriate for identifying the avascular area, with only one study showing that the diagnostic sensitivity of CH-EUS-FNA performed in the early phase was better than that of conventional EUS-FNA (Table 1). Nevertheless, the endosonographers are required to observe both the early and late phases for comprehensive assessment in actual clinical practice: the contrast effect of both phased should be taken into consideration when determining the portion of pancreatic masses to undergo EUS-FNA.

## 3. Avascular Areas on CH-EUS

Although EUS can detect pancreatic lesions, it is not accurate for the differential diagnosis of pancreatic masses. However, CH-EUS can characterize pancreatic masses by comparing the enhancement between solid pancreatic lesions and surrounding pancreatic tissues [6,17,18]. Four contrast patterns of internal blood flow define pancreatic solid lesions: iso-enhancement in mass-forming pancreatitis, hypo-enhancement in pancreatic adenocarcinoma, hyperenhancement in pancreatic neuroendocrine tumors, and nonenhancement in necrosis [4]. The typical CH-EUS image of pancreatic cancer is characterized by hypo-enhancement of most of the tumor, with a non-enhancing area (i.e., an avascular area) detected in part of the tumor. Numata et al. compared transabdominal contrast-enhanced ultrasound images of pancreatic cancer with pathological images of resected specimens and found that tumors with poor contrast enhancement contained a greater number of necrotic and fibrotic cells [19]. Furthermore, it is known that pancreatic neuroendocrine neoplasms with hypo-enhancement on CH-EUS have lower vessel density and greater fibrosis [20]. These reports also support the idea that CH-EUS-FNA helps to puncture hot spots in the tumor. Kamata et al. reported that the sensitivity of EUS-FNA for the diagnosis of pancreatic adenocarcinoma was significantly lower in cases with avascular areas than in those without (72.9% vs. 94.3%) [21]. One case report showed that EUS-FNA specimens from the avascular area were not suitable for detecting malignancy, whereas specimens obtained from the vascular area showed malignant findings [22]. In this report, EUS-FNA samples obtained from the avascular area was necrosis by pathological evaluations. This suggests that the avascular area on CH-EUS represents mostly necrotic or fibrotic tissue, and CH-EUS-FNA can help avoid this area. In certain cases, an avascular area on CH-EUS is recognized on normal EUS as an area within the tumor that is hypoechoic compared with the surrounding lesion (Figure 1). However, avascular areas identified on CH-EUS may often not be detected on normal EUS (Figure 2). A difference in the degree of echogenicity within the tumor on EUS is not necessarily identified as an avascular area on CH-EUS. Thus, the area imaged as the avascular area in CH-EUS may be an inappropriate specimen. Therefore, in order to obtain an appropriate specimen for diagnosis, it is necessary to puncture the area, avoiding the avascular area. Although no reports specifically describe the technique to avoid the avascular area during CH-EUS-FNA, it would be better to puncture the areas with blood flow in the tumor, changing the angle of the EUS-FNA needle using forceps to raise up or down the angle of the endoscopy.

## 4. Diagnostic Capability of CH-EUS-FNA

Three meta-analyses that included a large number of studies reported that CH-EUS shows superior performance for the diagnosis of solid masses [14,23,24]. Eight reports evaluated the pathological diagnostic performance of CH-EUS-FNA for pancreatic masses (Table 1) [5,6,7,8,9,10,11,12], with these including six prospective studies and two retrospective ones, although two of the prospective studies were single-arm designs. The number of patients in these studies ranged from 35 to 225. Most studies did not describe the number of cases with avascular areas, but Sugimoto et al. reported that 20 consecutive cases evaluated with CH-EUS-FNA had avascular areas, and Itonaga et al. reported that 41.5% (34/93) of cases had an avascular area. However, the definition of avascular area was ambiguous in these two studies. Previously, Kamata et al. defined tumors with an avascular area as those with a non-enhancing area ≥5 mm on CH-EUS, and reported that 16.4% (48/292) of pancreatic masses had an avascular area [21]. The variation in the proportion of cases with an avascular area could be due to differences in the definition. In the eight studies listed in Table 1, CH-EUS-FNA was performed by expert endosonographers using a 22-gauge EUS-FNA needle, whereas data obtained using an EUS-fine needle biopsy (FNB) needle are lacking. Regarding the puncture site, most studies reported avoiding avascular areas, and three studies reported detecting a hypo-enhanced area. Puncturing the hypo-enhanced area, which indicates pancreatic cancer, is reasonable, especially in pancreatic masses without an avascular area. Two prospective studies (Napoleon et al., 2010 and Gincul et al., 2014) demonstrated the feasibility of CH-EUS-FNA in a single-arm study [5,6], whereas six studies compared the diagnostic accuracy of CH-EUS-FNA and EUS-FNA. Among these studies, two performed both CH-EUS-FNA and EUS-FNA in the same patients (Seicean et al., 2015 and Itonaga et al., 2020) [9,12]. The sensitivity of CH-EUS-FNA ranged from 79% to 96%, and its specificity from 90% to 100%. Six studies showed that the sensitivity of CH-EUS-FNA was higher than that of EUS-FNA, but only one study showed that the difference was significant (*p* = 0.003; Table 1). However, in this study showing a significant difference, the sensitivity of normal EUS-FNA was particularly low at 68.8%, which could be attributed to the fact that a single pass was used to compare the diagnostic performance of the two methods, rather than the multiple passes used in other studies. Moreover, the first pass was performed using EUS-FNA and the second pass using early-phase CH-EUS with the avascular area confirmed. The specimen obtained by single pass was used for evaluation, and EUS-FNA was performed prior to CH-EUS-FNA. In summary, the added value of CH-EUS-FNA in comparison with EUS-FNA remains unclear, and further studies are needed.

The precision of EUS-FNA is considered to be contingent upon the proficiency of the endosonographers. Additionally, the assessment of pancreatic lesions through CH-EUS and the detection of the avascular area are also subject to their examination skills. Thus, standardization of the procedures and diagnostic proficiency is imperative to gauge the impact of ultrasound contrast agents in CH-EUS. In addition, improvements in examination equipment such as endoscopes and EUS-FNA needles may also have an impact on CH-EUS-FNA in the future.

## 5. Detection of Subtle Lesions

In some cases, CH-EUS depicts pancreatic tumors that are difficult to identify on normal EUS. This may be due to the fact that the contrast effect achieves clearer margins to pancreatic tumors.

Kitano et al. performed CH-EUS on 277 patients with pancreatic tumors, and among them, six cases of ductal carcinoma were depicted on CH-EUS but not on EUS [18]. Kamata et al. evaluated the utility of CH-EUS for the surveillance of remnant pancreas after surgery for intraductal papillary mucinous neoplasm (IPMN) [25]. One hundred and thirty-four patients were followed-up for a median of 29 months, and CH-EUS was useful to identify two cases of small IPMN concomitant with pancreatic ductal adenocarcinoma that developed during the follow-up period. Fusaroli et al. also mentioned that CH-EUS was useful for detecting pancreatic masses, especially in patients with a bile duct stent or chronic pancreatitis [17].

Thus, there are some pancreatic masses that are detected only on CH-EUS, and not on ordinary EUS. The prognosis for advanced pancreatic cancer is poor, and early detection and treatment are critical. Given the difficulty in detecting certain lesions through ordinary EUS, it is also important to conduct a comprehensive screening of the pancreas using CH-EUS rather than just limiting the examination to lesions that are detectable through ordinary EUS.

CH-EUS also allows for clearer imaging of the tumor margins of lesions, which leads to reliable puncture of the tumor on EUS-FNA.

## 6. Risks Related to CH-EUS-FNA

There are few risks associated with the use of ultrasound contrast agents or performing EUS-FNA under CH-EUS guidance. The color Doppler mode is generally used to identify vessels on the puncture line during EUS-FNA, although this method is often limited by blooming artifact [26]. There are no reports indicating that EUS-FNA with CH-EUS guidance increases the incidence of adverse events; however, it is more difficult to identify vessels using CH-EUS than using color Doppler mode. If CH-EUS detects a pancreatic mass without an avascular area, it might be possible to switch to normal EUS-FNA. The blooming artifact associated with the color Doppler mode occurs more frequently after the administration of ultrasound contrast medium. Another limitation of CH-EUS-FNA is that the tip of the EUS-FNA needle can be difficult to visualize in deep areas away from the EUS probe. The side effects of the ultrasound contrast medium are rarely a clinical problem [27,28]; mild allergy-like symptoms can occur, but there are no reports of serious events [27,28].

## 7. Future Perspectives

Although several studies have described the utility of CH-EUS-FNA, only one study demonstrated the value of CH-EUS-FNA for improving diagnostic performance. It is therefore important to clarify the specific cases that would benefit from CH-EUS-FNA.

CH-EUS-FNA may be particularly valuable in cases in which it is difficult to obtain an accurate diagnosis—for example, in cases in which EUS-FNA is performed with a small-gauge needle, by beginners, or using a single puncture. Recently developed EUS-FNB needles, such as fork-tip or shark-core needles, have improved the quality and quantity of pancreatic specimens collected [29,30], and improvements in the diagnostic performance of EUS-FNA may decrease the impact of CH-EUS guidance for EUS-FNA in the future. However, precision medicine is attracting increased attention, and the ability to perform oncogene panel testing on EUS-FNA specimens is essential [31]. For such testing, it is important to collect cancer tissues from a site with a high cell count. In this sense, CH-EUS-FNA is still expected to play an important role in pancreatic cancer tissue collection.

Pancreatic cancer is likely to metastasize to the lymph nodes and its staging depends on the presence of lymph node metastasis [32]. However, the correct diagnosis of lymph node metastases remains challenging [33]. Kurita et al. compared the diagnostic performance of EUS-FNA and PET-CT for para-aortic lymph node metastasis in pancreatobiliary cancer [34]. Fifty-two patients had enlarged para-aortic lymph nodes, and postoperative diagnosis of lymph node metastasis was made in 21 patients (40.4%). In this study, EUS-FNA showed superior accuracy to PET-CT in the diagnosis of malignancy in lymph nodes (95.2% vs. 57.1%).

If there are many enlarged lymph nodes, it is hard to perform EUS-FNA on all of them in clinical practice. Therefore, it is useful to estimate the malignancy of enlarged lymph nodes by CH-EUS in order to determine the optimum target for EUS-FNA.

Miyata et al. examined enlarged intra-abdominal lymph nodes associated with pancreatobiliary cancers in 143 patients. Heterogeneous enhancement on CH-EUS was a sign of malignancy, and CH-EUS had a sensitivity of 83% and specificity of 91% [35]. Recently, a meta-analysis including 336 cases in four studies showed that CH-EUS had a sensitivity of 82.1% and specificity of 90.7% for diagnosing malignant lymph nodes [36].

In recent years, EUS-BD has become a popular method of bile duct drainage. Several cases were reported in which CH-EUS improved the visibility of bile ducts that were poorly viewed on normal EUS because of debris or sludge during EUS-BD procedures [37,38].

Minaga et al. also mentioned that CH-EUS was helpful for EUS-guided cyst drainage for pancreatic pseudocyst or walled-off necrosis (WON) [39]. CH-EUS allows for real-time evaluation of blood flow and can assist in distinguishing WON from other luminal organs, which leads to a safer procedure in EUS-guided drainage.

The utilization and development of artificial intelligence (AI) in the medical field is ongoing. Numerous studies examining the diagnostic efficacy of AI in ordinary EUS for pancreatic cancer have yielded favorable outcomes [40,41,42,43]. AI-based examination of CH-EUS images also has been explored, although not applied to the analysis of pancreatic lesions. Tanaka et al. applied AI to differentiate gastrointestinal stromal tumors and leiomyomas in the identification of gastric submucosal tumors, yielding favorable results [44]. If AI could accurately differentiate gastric submucosal tumors through CH-EUS, it is within the realm of possibility that it could also distinguish pancreatic masses. Research on AI-based analysis and diagnosis of CH-EUS images for pancreatic lesions is also expected to advance in the future.

Thus, CH-EUS might be useful for performing EUS-FNA or EUS-guided drainage for pancreaticobiliary lesions, and further research including large-scale randomized clinical trials is desired. As mentioned above, there are few side effects associated with ultrasound contrast agents, and the benefits might far exceed the risks. In addition, the administration is easy because an intravenous route for sedation is generally secured during EUS examinations. Nevertheless, CH-EUS is still not covered by insurance in any of the regions, and CH-EUS is not allowed to be performed in general practice. Further evidence on the usefulness of CH-EUS is needed to achieve insurance approval for CH-EUS.

## Figures and Tables

**Figure 1 diagnostics-13-01039-f001:**
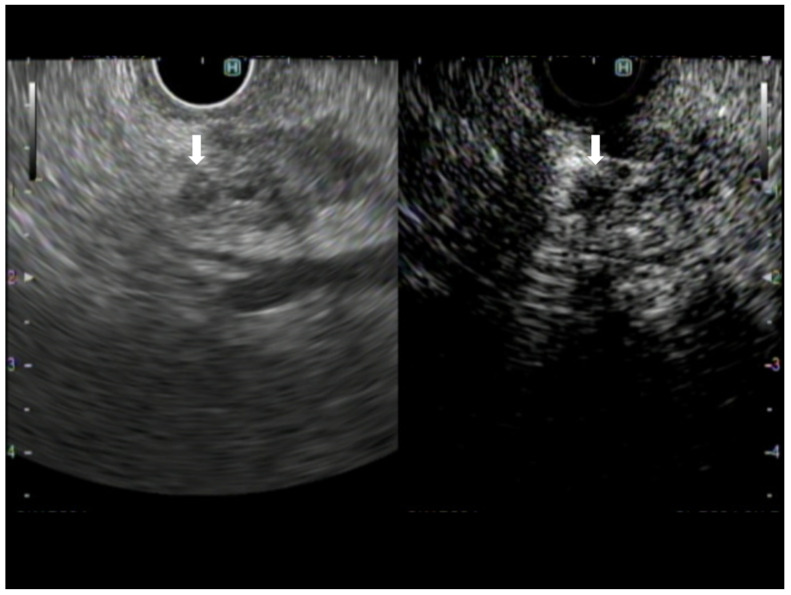
A case of pancreatic adenocarcinoma. The image on the left shows B-mode EUS, and the image on the right shows CH-EUS. The avascular area observed in the CH-EUS image is recognized on B-mode EUS as a hypoechoic area compared with the surrounding lesion within the tumor (arrow).

**Figure 2 diagnostics-13-01039-f002:**
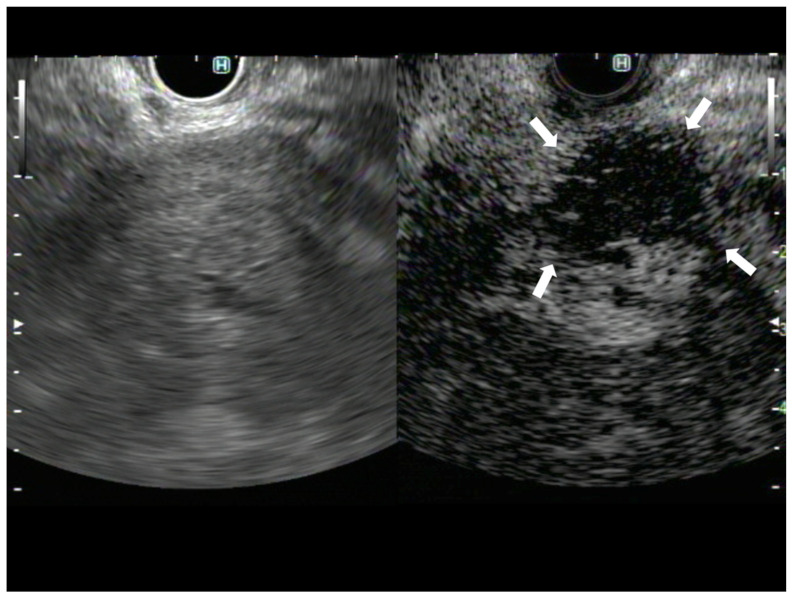
A case of pancreatic adenocarcinoma. The image on the left shows B-mode EUS, and the image on the right shows CH-EUS. The avascular area observed in the CH-EUS image was not identified on B-mode EUS (arrows).

**Table 1 diagnostics-13-01039-t001:** Studies on CH-EUS-FNA for pancreatic masses.

Reference	Study Design	Number of Patients, n		Number of Cases with Avascular Area, n (%)	Experience of Endosonographer	Needle-Gauge	Contrast Agent	Timing of CH-EUS-FNA	Target of CH-EUS-FNA	Outcome Measure					
		CH-EUS-FNA	EUS-FNA							CH-EUS-FNA		EUS-FNA		Statistical Difference	
										Sensitivity	Specificity	Sensitivity	Specificity	Sensitivity	Specificity
Napoleon et al., 2010 [5]	Prospective	35	0	No data	No data	22	Sulphur hexafluoride microbubbles	Late phase	No data	79.0%	100%	No data	No data	No data	No data
Gincul et al., 2014 [6]	Prospective	100	0	No data	No data	22	Sulphur hexafluoride microbubbles	Late phase	Hypo-enhanced area	96.0%	94.0%	No data	No data	No data	No data
Hou et al., 2015 [7]	Retrospective	58	105	No data	No data	22	Sulphur hexafluoride microbubbles	No data	Hypo-enhanced area	81.6%	100%	70.8%	100%	NS	
Sugimoto et al., 2015 [8]	Prospective	20	20	20/20 (100%)	Less than 100 EUS-FNA	22	Perflubutane microspheres	Late phase	Avoiding avascular area	90.0%	No data	85.0%	No data	0.500	No data
Seicean et al., 2015 [9]	Prospective	51 (both were performed on the same patients)		No data	No data	22	Sulphur hexafluoride microbubbles	Late phase	Avoiding avascular area	82.9%	100%	73.2%	100%	NS	
Facciorusso et al., 2020 [10]	Retrospective	103	103	No data	20 years of experience	22	Sulphur hexafluoride microbubbles	No data	Hypo-enhanced area	87.6%	100%	80.00%	100%	0.180	1.000
Seicean et al., 2020 [11]	Prospective	75	75	No data	Over 7000 EUS-FNA and 500 CH-EUS	22	Sulphur hexafluoride microbubbles	Late phase	Avoiding avascular area	87.6%	100%	85.5%	100%	NS	
Itonaga et al., 2020 [12]	Prospective	93 (both were performed on the same patients)		34/93 (41.5%)	Over 300 EUS-FNA	22	Perflubutane microspheres	Early phase	Avoiding avascular area	84.9%	100%	68.8%	100%	0.003	NS

CH-EUS-FNA, contrast-enhanced harmonic endoscopic ultrasound-guided fine-needle aspiration; EUS-FNA, endoscopic ultrasound-guided fine-needle aspiration; NS, not significant.

## Data Availability

Data sharing not applicable.

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
