# Peer review of "Contrast-Enhanced Harmonic Endoscopic Ultrasound-Guided Puncture for the Patients with Pancreatic Masses"

_diagnostics, 2023, doi:10.3390/diagnostics13061039_

Round 1

Reviewer 1 Report (Previous Reviewer 2)

The manuscript has been revised well. However, some data should be addressed to show the excellence of this article.

1.      I think it would be better if you could compare and present the actual pathological findings and contrast enhanced EUS images of resected cases. Authors should demonstrate in actual specimens that there are no viable cells in the avascular area.

Author Response

Answer:  We appreciate your important suggestion. I had received that suggestion from reviwer2 during the last revise. To our knowledge, there is no report on comparison between avascular area on CH-EUS and pathological findings in the resected cases with pancreatic cancer. However, it is known that severe fibrosis and necrosis were seen in pancreatic cancers with avascular areas on contrast-enhanced ultrasonography in resected specimens as mentioned in the section of “Avascular areas on CH-EUS” of the previous manuscript. Furthermore, we have reported that pancreatic neuroendocrine neoplasms with hypo-enhancement on CH-EUS had lower vessel density and greater fibrosis [R. Ishikawa, et al. Dig Endosc, 2021. 33(5): p.829-839.]. We would like to verify this in our next study comparing avascular area on CH-EUS with pathological findings.

Reviewer 2 Report (Previous Reviewer 1)

It is OK.

Author Response

Answer: We appreciate your advice which has lead us to a better article.

Round 2

Reviewer 1 Report (Previous Reviewer 2)

This paper is an important contribution and I recommend that it be accepted for publication.

This manuscript is a resubmission of an earlier submission. The following is a list of the peer review reports and author responses from that submission.

Round 1

Reviewer 1 Report

In the abstract and section 2, “thereby improving the diagnostic performance of EUS-FNA” should not be described, because little evidence of the improvement of EUS-FNA by using contrast medium is shown in the article.

Third section of “The CH-EUS FNA technique should be placed before 2nd section of” Avascular area on CH-EUS”.

“Sonazoid” , “Sono Vue” , and “ Deffinity” are brand names.  For example, “perflubutane microspheres” is a chemical name of “Sonazoid”.

“Sonazoid” is not approved for the pancreatobiliary masses in Japan. Are there any countries where “ Sonazoid” or other contrast media such as “Sono Vue” and “ Deffinity” are approved for the pancreatic tumor ?

Are there any differences between “Sonazoid” and other media about methods to perform CH-EUS and obtained images? 

Are there any differences between 3 meta-analyses and this article? 

How to select the literatures shown in table 1?

Although the title of the manuscript is “Contrast-enhanced harmonic endoscopic ultrasound-guided fine-needle aspiration for pancreatic masses”, the evaluations of lymph node metastases and UES-BD EUS-guide cyst drainage are also discussed in contrast to the title.

Author Response

In the abstract and section 2, “thereby improving the diagnostic performance of EUS-FNA” should not be described, because little evidence of the improvement of EUS-FNA by using contrast medium is shown in the article.

Answer: We appreciate your important suggestion. Accordingly, we deleted the following sentence “thereby improving the diagnostic performance of EUS-FNA” in the abstract and the section of “Avascular areas of CH-EUS” of the revised manuscript.

Third section of “The CH-EUS FNA technique should be placed before 2nd section of “Avascular area on CH-EUS”.

Answer: We thank you for the careful review of the manuscript. We moved the section of “The CH-EUS FNA technique” before 2nd section of “Avascular area on CH-EUS” in the revised manuscript.

“Sonazoid”, “Sono Vue”, and “Deffinity” are brand names.  For example, “perflubutane microspheres” is a chemical name of “Sonazoid”.

Answer: Thank you very much for pointing it out. We changed brand names of contrast agents to their chemical names in the revised manuscript.

“Sonazoid” is not approved for the pancreatobiliary masses in Japan. Are there any countries where “Sonazoid” or other contrast media such as “SonoVue” and “Deffinity” are approved for the pancreatic tumor?

Answer: Thank you very much for these excellent comments. To our knowledge, none of the contrast agents are approved for the diagnosis of pancreatic masses in any countries.

Are there any differences between “Sonazoid” and other media about methods to perform CH-EUS and obtained images? 

Answer: Thank you very much for your nice comment. Unlike other contrast medias, Sonazoid has the advantage of obtaining a Kupffer image. However, since there are no Kupffer cells in the pancreas, the significance of this advantage in the diagnosis of pancreatic tumors is not presently known. Thus, any second-generation ultrasound contrast agents can be used for the diagnosis of pancreatic tumors. We added this to the section of “The CH-EUS-FNA technique” of the revised manuscript.

Are there any differences between 3 meta-analyses and this article? 

Answer: Thank you very much for your important question. Three meta-analyses mainly examined the diagnostic performance of CH-EUS for the differential diagnosis of pancreatic masses. By contrast, this article reviewed with respect to CH-EUS-guided puncture including CH-EUS-FNA in the patients with pancreatic masses.

How to select the literatures shown in table 1?

Answer: We appreciate helpful suggestion. In selecting the literatures, the the following search terms were used: contrast (title or abstract) OR contrast-enhanced (title or abstract) OR contrast-enhanced harmonic (title or abstract) OR CE-EUS (title or abstract) OR CH-EUS (title or abstract) OR CEH-EUS (title or abstract) AND endoscopic ultrasound (title or abstract) OR EUS (title or abstract) OR endosonography (title or abstractor MeSH terms) OR endoscopic ultrasonography (title or abstract) AND FNA (title or abstract) OR FNB (title or abstract) OR fine needle aspiration (title or abstract) OR fine needle biopsy (title or abstract) OR sampling (title or abstract).Then, after sequential screening of abstracts and texts, eight studies were determined. We added this in the section of “The CH-EUS-FNA technique” of the revised manuscript.

Although the title of the manuscript is “Contrast-enhanced harmonic endoscopic ultrasound-guided fine-needle aspiration for pancreatic masses”, the evaluations of lymph node metastases and UES-BD EUS-guide cyst drainage are also discussed in contrast to the title.

Answer: Thank you very much for nice suggestion. We changed the previous title to “Contrast-enhanced harmonic endoscopic ultrasound-guided puncture for the patients with pancreatic masses”.

Reviewer 2 Report

To author

In this review, Otsuka et al showed the feasibility of contrast-enhanced harmonic endoscopic ultrasonography (CE-EUS) in EUS-guided fine-needle aspiration for pancreatic masses. This review is useful and important because the characteristics of this procedure were shown. However, some data should be addressed to show the excellence of this article.

1.      This review requires a detailed explanation of the CE-EUS-FNA technique to avoid avascular area.

2.      The authors should demonstrate the absence of tumor cells in the avascular area on histopathological examination in each report.

3.      The authors should explain the late phase in the pancreas, in which, unlike the liver, there are no Kupffer cells.

Author Response

1. This review requires a detailed explanation of the CE-EUS-FNA technique to avoid avascular area.

Answer: We thank you very much for the positive comment. Although no reports specifically describe the technique to avoid avascular area, it would be better to puncture the areas with blood flow in the tumor changing the angle of the EUS-FNA needle using forceps raising or up and dawn angle of the endoscopy. We added this to the section of “Avascular areas on CH-EUS” of the revised manuscript.

2. The authors should demonstrate the absence of tumor cells in the avascular area on histopathological examination in each report.

Answer: Thank you very much for the insightful comment. To our knowledge, one case report showed that EUS-FNA samples obtained from the avascular area was necrosis by pathological evaluations [Ueda, K et al. Dig Endosc, 2013 25: 631]. We have added this to the section of "Avascular areas on CH-EUS" of the revised manuscript.

3. The authors should explain the late phase in the pancreas, in which, unlike the liver, there are no Kupffer cells.

Answer: Thank you very much for important comment. Sonazoid allows contrast-enhanced ultrasound evaluations at early phase, late phase, and Kupffer phase. The early, late, and Kupffer phases are defined as 10-30 seconds, 30-120 seconds, and 10 minutes after injection of the contrast agent, respectively. As you pointed it out, there are no Kupffer cells in the pancreas; therefore, early and late phases are used for CH-EUS evaluations for pancreatic lesions. We added this to the section of “The CH-EUS-FNA technique” of the revised manuscript.

Round 2

Reviewer 1 Report

All comments I and another reviewer  provided   were appropriately corrected.

Author Response

We appreciate your advice which has lead us to a better article.

Reviewer 2 Report

The manuscript has been revised well. However, some data should be addressed to show the excellence of this article.

1.      I think it would be better if you could compare and present the actual pathological findings and contrast enhanced EUS images of resected cases.

Author Response

1. I think it would be better if you could compare and present the actual pathological findings and contrast enhanced EUS images of resected cases.

Answer: We appreciate your important suggestion. Although not reported by CH-EUS, it is known that severe fibrosis and necrosis were seen in pancreatic cancers with avascular areas on contrast-enhanced ultrasonography in resected specimens as mentioned in the section of “Avascular areas on CH-EUS” of the previous manuscript. Furthermore, we have reported that pancreatic neuroendocrine neoplasms with hypo-enhancement on CH-EUS had lower vessel density and greater fibrosis [R. Ishikawa, et al. Dig Endosc, 2021. 33(5): p.829-839.]. These reports also support the idea that CH-EUS-FNA helps to puncture hot spots in the tumor. We added this to the section of “Avascular areas on CH-EUS” of the revised manuscript. To our knowledge, there is no report on comparison between avascular area on CH-EUS and pathological findings in the resected cases with pancreatic cancer, and we would like to verify this in our next study.

Round 3

Reviewer 2 Report

This manuscript has been revised well. I think this manuscript is acceptable.